# Deciphering the Effects of the PYCR Family on Cell Function, Prognostic Value, Immune Infiltration in ccRCC and Pan-Cancer

**DOI:** 10.3390/ijms25158096

**Published:** 2024-07-25

**Authors:** Hongquan Chen, Qing Chen, Jinyang Chen, Yazhen Mao, Lidi Duan, Dongjie Ye, Wenxiu Cheng, Jiaxi Chen, Xinrong Gao, Renxi Lin, Weibin Lin, Mingfang Zhang, Yuanlin Qi

**Affiliations:** 1School of Basic Medical Sciences, Fujian Medical University, Fuzhou 350122, China; 13843202849@163.com (H.C.); cq1392791987@fjmu.edu.cn (Q.C.); myz@fjmu.edu.cn (Y.M.); 13663730251@163.com (L.D.); yedj@fjmu.edu.cn (D.Y.); wenxiucheng@fjmu.edu.cn (W.C.); cjx123@fjmu.edu.cn (J.C.); 2210110046@fjmu.edu.cn (X.G.); linrenxi89@fjmu.edu.cn (R.L.); weibinlin@fjmu.edu.cn (W.L.); 2College of Computer and Cyber Security, Fujian Normal University, Fuzhou 350009, China; a302850047@gmail.com

**Keywords:** pyrroline-5-carboxylate reductase (PYCR), pan-cancer analysis, prognostic model, kidney renal clear cell carcinoma (KIRC), Pathomics, mTOR

## Abstract

Pyrroline-5-carboxylate reductase (PYCR) is pivotal in converting pyrroline-5-carboxylate (P5C) to proline, the final step in proline synthesis. Three isoforms, PYCR1, PYCR2, and PYCR3, existed and played significant regulatory roles in tumor initiation and progression. In this study, we first assessed the molecular and immune characteristics of PYCRs by a pan-cancer analysis, especially focusing on their prognostic relevance. Then, a kidney renal clear cell carcinoma (KIRC)-specific prognostic model was established, incorporating pathomics features to enhance predictive capabilities. The biological functions and regulatory mechanisms of PYCR1 and PYCR2 were investigated by in vitro experiments in renal cancer cells. The PYCRs’ expressions were elevated in diverse tumors, correlating with unfavorable clinical outcomes. PYCRs were enriched in cancer signaling pathways, significantly correlating with immune cell infiltration, tumor mutation burden (TMB), and microsatellite instability (MSI). In KIRC, a prognostic model based on PYCR1 and PYCR2 was independently validated statistically. Leveraging features from H&E-stained images, a pathomics feature model reliably predicted patient prognosis. In vitro experiments demonstrated that PYCR1 and PYCR2 enhanced the proliferation and migration of renal carcinoma cells by activating the mTOR pathway, at least in part. This study underscores PYCRs’ pivotal role in various tumors, positioning them as potential prognostic biomarkers and therapeutic targets, particularly in malignancies like KIRC. The findings emphasize the need for a broader exploration of PYCRs’ implications in pan-cancer contexts.

## 1. Introduction

Cancer stands as a predominant cause of mortality across the globe [1], and the intricate nature of the tumor microenvironment presents formidable challenges to cancer treatment. Despite ongoing advancements in therapeutic modalities, the worldwide incidence and mortality rates of cancer continue to escalate rapidly [2]. Given the intricate processes underlying cancer initiation, a comprehensive exploration of the biological characteristics of genes assumes paramount importance for clinical treatment and prognosis prediction. Through a systematic analysis of cancer gene expression and variations, leveraging the Cancer Genome Atlas (TCGA) database [3], we can delve into the exploration of new prognostic biomarkers and therapeutic targets. This approach aims to unravel the intricate relationship between specific genes and immunotherapy, potentially paving the way for innovative strategies in cancer treatment.

In recent years, the interconversion between proline, glutamate, and ornithine has emerged as a focal point in the regulation of tumor cell growth in the field of cancer metabolism. ∆1-pyrroline-5-carboxylate (P5C) serves as the direct precursor of proline and is also a product of proline degradation. The conversion of P5C to proline is catalyzed by PYCR, the final enzyme in proline biosynthesis [4,5]. Three isoforms of PYCR have been identified—PYCR1, PYCR2, and PYCR3/PYCRL. Both PYCR1 and PYCR2 are located in the mitochondria, sharing 84% structural homology. They function similarly in the final step of the glutamate-P5C-proline pathway, exhibiting a preference for NADH as a cofactor [6]. PYCR3, on the other hand, lacks 40 amino acids at the C-terminus and is mainly situated in the cytoplasm. Additionally, PYCR3 prefers utilizing NADPH as a cofactor to catalyze the production of proline from ornithine [7]. Recent studies indicate that PYCRs are upregulated in various tumors and are associated with the development of certain cancers. For example, downregulating PYCR1 can inhibit the growth of LUAD cells by impacting the JAK/STAT signaling pathway [8]. In vitro and in vivo studies have demonstrated the ability to suppress the proliferation, invasion, epithelial-mesenchymal transition (EMT), and metastasis of hepatocellular carcinoma [9]. PYCRs can also modulate the STAT3-mediated p38 MAPK and NF-κB signaling pathways in colorectal cancer to inhibit proliferation, drug resistance, and the EMT signaling pathway [10]. Knocking out PYCR2 inhibits the proliferation, migration, and invasion of colon cancer [10,11] and also influences the development of melanoma, bladder cancer, nasopharyngeal carcinoma, gastric cancer, and other tumors [12,13,14,15]. However, current research on PYCRs in a pan-cancer context is lacking, and there is a dearth of studies utilizing PYCRs to predict tumor prognosis.

Significant progress has been achieved in the application of artificial intelligence algorithms in the medical field, particularly in the diagnosis and treatment of cancer, with notable success in recent years [16,17]. Machine learning and deep learning are widely utilized in the medical image processing of various cancers, such as breast cancer, lung adenocarcinoma, and neurogenic tumors [18,19,20]. These technologies enable high-throughput analysis of medical images, allowing artificial intelligence to extract extensive information crucial for precision medicine [19]. Pathomics, employing artificial intelligence methods, extract high-throughput features from pathology images stained with Hematoxylin and Eosin (H&E). By combining pathological morphological information with genetic data, this approach reveals imperceptible insights and uncovers patterns that are challenging to discern through subjective observation alone [21]. By integrating microscopic-level image information at the cellular level, there is promising potential to significantly enhance the predictive efficiency of disease diagnosis.

In this study, we first analyzed the expression patterns and prognostic value of PYCRs in a pan-cancer context based on the TCGA database. Subsequently, we examined the correlation between PYCR expression and tumor mutation burden (TMB), microsatellite instability (MSI), tumor immune infiltration, and potential biological pathways. Risk score models were constructed using PYCR1 and PYCR2 specifically for KIRC. We investigated the relationship between the risk score models and the clinical prognosis of overall survival in KIRC, followed by an analysis of the molecular and immune features of the risk score models. Furthermore, we utilized features extracted from H&E-stained images of KIRC to build a pathomics feature model predicting the prognosis risk score model we designed. Univariate and multivariate Cox regression analyses were employed to confirm the stability of the pathomics feature model in predicting the prognosis of KIRC patients. Finally, we explored the impact of PYCR1 and PYCR2 on cell growth and migration in renal cancer cells, revealing that PYCR1 and PYCR2 may exert their effects on renal cancer by activating the mTOR signaling pathway.

## 2. Results

### 2.1. Analysis of PYCRs in Pan-Cancer

We analyzed the mRNA expression levels of PYCRs in various cancer types and corresponding normal tissues using TCGA data. PYCR1 (Figure 1A), PYCR2 (Appendix A), and PYCR3 (Appendix A) were upregulated in 18, 15, and 15 types of tumors, respectively, while PYCR3 showed decreased expression in KICH, KIRC, and THCA. Single-factor Cox regression analysis revealed that PYCR1 was a high-risk gene for overall survival (OS) and progression-free survival (PFS) in eight cancer types (Figure 1B). Similarly, PYCR2 and PYCR3 were identified as high-risk genes in multiple cancer types for OS and PFS (Appendix A).

GSEA enrichment analysis of PYCRs showed that in GBM, PRAD, and SKCM, PYCR1 negatively regulates TNF-α signaling and IL6-JAK-STAT3 signaling, while in ACC, KIRC, KIRP, and THCA, PYCR1 positively regulates G2M checkpoint, EMT, and E2F targets (Figure 1C). PYCR2 exhibited similar enrichment patterns in pan-cancer (Appendix A). For PYCR3, it negatively regulates TNF-α signaling and IL6-JAK-STAT3 signaling in LGG, PAAD, PRAD, and STAD, while positively regulating G2M checkpoint and E2F targets in ACC, CHOL, ESCA, PAAD, and STAD (Appendix A).

We analyzed the relationship between the expression of PYCRs and tumor mutational burden (TMB) as well as microsatellite instability (MSI). The expression of PYCR1 was positively correlated with TMB in 18 cancer types and positively correlated with MSI in 9 cancer types, while it was negatively correlated with MSI in READ (Figure 1D,E). The expression of PYCR2 was positively correlated with TMB in 6 cancer types and positively correlated with MSI in eight cancer types but negatively correlated with TMB in four tumor types and MSI in three tumor types (Appendix A). PYCR3 expression was positively correlated with TMB in 11 cancer types and positively correlated with MSI in eight cancer types, while negatively correlated with TMB in COAD and MSI in both COAD and READ (Appendix A).

The results of immune cell infiltration analysis showed that in 8 types of tumors, including STAD and KIRC, the expression of PYCR1 was positively correlated with the infiltration of T cells CD4 memory activated. In six types of tumors, including LUSC and LUAD, the expression of PYCR1 was positively correlated with the infiltration of T cells CD8 (Figure 1F). In six types of tumors, including STAD and SARC, the expression of PYCR2 was positively correlated with the infiltration of T cells CD8 (Appendix A). In 11 types of tumors, including LUAD and KIRC, the expression of PYCR3 was positively correlated with the infiltration of T cells CD8 (Appendix A).

### 2.2. Construction of OS-Related Prognostic Risk Score Model for PYCRs in KIRC

In the pan-cancer prognosis analysis, we observed a significant correlation between the entire PYCR family genes and the prognosis of KIRC (Appendix A). Utilizing the transcriptomic expression data of PYCRs and OS data in KIRC, we conducted LASSO regression analysis and identified PYCR1 and PYCR2 as two genes for constructing the prognostic risk model (Appendix A). Subsequently, a multifactor Cox regression analysis was performed to further confirm that PYCR1 and PYCR2 could be used to construct the risk score model (Appendix A). Using the median value of the risk score as the threshold, we divided the training and internal validation cohorts into high and low-risk groups. In the training cohort, the high-risk group showed significantly reduced OS (Figure 2A), and the time-related ROC curve results indicated that the AUC of the risk score model for 1-year, 3-year, and 5-year OS were 0.725, 0.692, and 0.711, respectively (Figure 2B). Additionally, we ranked the risk scores of each patient in the training cohort, with higher scores indicating shorter survival times and fewer surviving patients. The heatmap described the expression patterns of PYCR1 and PYCR2 in the high-risk and low-risk groups (Figure 2C). The predictive ability of this risk score model was validated in the internal validation cohort (Figure 2D–F) and the external validation cohort (Figure 2G–I), yielding results similar to the training cohort. This suggests that the risk score model we constructed exhibits high accuracy in predicting OS for KIRC patients. The results of univariate and multivariate Cox regression analyses assessing the impact of clinical-pathological factors and the risk score from the prognostic model on the OS of KIRC patients indicated that the risk score from the model is an independent prognostic factor for KIRC patients (Figure 2G).

### 2.3. Molecular and Immune Features of Risk Score Model Subgroups

The results of GSEA enrichment analysis demonstrated significant enrichment of immune-related pathways in the GO gene set in the high-risk score group of KIRC, while the low-risk score group exhibited significant enrichment in spliceosome assembly-related pathways and pathways related to compound and metal ion reactions (Figure 3A). In the Hallmark gene set, the high-risk groups of KIRC showed significant enrichment in pathways such as E2F Targets, G2/M Checkpoint, and EMT, while the low-risk score group of KIRC exhibited significant enrichment in pathways related to bile, fat metabolism, and others (Figure 3B). We conducted an analysis of gene mutations in the subgroups of the risk score model. In KIRC, the mutation rates in the high-risk group were slightly higher than those in the low-risk group. The most frequent mutation type was missense mutation, followed by frameshift deletion mutation. VHL, PBRM1, and TTN had mutation rates exceeding 10% in both high and low-risk groups of KIRC (Figure 3C,D). We also explored the correlation between the risk score and tumor mutation burden (TMB), and the results showed a positive correlation between TMB and risk score in KIRC (Figure 3E). The infiltration of immune cells was compared between subgroups of the risk score model. In KIRC, Plasma cells, T cells CD8, T cells CD4 memory activated, T cells follicular helper, T cells regulatory (Tregs), NK cells activated, and Macrophages M0 exhibited higher infiltration abundance in the high-risk score group. T cells CD4 memory resting, NK cells resting, Monocytes, Macrophages M1, Macrophages M2, and Mast cells resting showed higher infiltration abundance in the low-risk score group (Figure 3F). This suggests that the risk score model we constructed can be used to reflect the state of the tumor microenvironment in KIRC.

### 2.4. Pathomics Feature Model Predicts Prognostic Risk Score Model

Through feature selection using mRMR and RFE algorithms, we ultimately selected five pathological features (Figure 4A). A pathomics feature model was constructed using the GBM algorithm, and Figure 4B displays the importance of the features selected in the GBM algorithm. The model’s performance was comprehensively evaluated, and the indicators in the training cohort included sensitivity, specificity, accuracy, positive predictive value (PPV), negative predictive value (NPV), and Brier score, which were 0.667, 0.857, 0.765, 0.815, 0.732, and 0.178, respectively. In the validation cohort, the indicators were 0.612, 0.876, 0.749, 0.822, 0.708, and 0.208 (Appendix A). In the training cohort, the AUC values of the ROC curve were 0.821, and the AUC values of the PR curve reached 0.841. Calibration curves showed good fitting between our predictions and actual gene levels. Decision curve analysis indicated that our pathomics feature model achieved the maximum net benefit at a threshold probability of 0.2–1, demonstrating clinical utility (Figure 4C). In the validation cohort, the AUC values of the ROC curve were 0.741, and the AUC values of the PR curve reached 0.736. Calibration curves showed good fitting between our predictions and actual gene levels. Decision curve analysis indicated that our pathomics feature model achieved the maximum net benefit at a threshold probability of 0.35–1, demonstrating clinical utility (Figure 4D). The prognostic risk score probability, Pathomics score, outputted by the pathomics feature model was higher in the high-risk group than in the low-risk group in both the training and the validation cohorts (Figure 4E).

Based on the median pathomics score of the GBM model, samples were divided into high-risk (*n* = 204) and low-risk groups (*n* = 203) for prognostic analysis of KIRC patients. The Kaplan-Meier curve showed that patients in the high-risk group had a poorer prognosis (Figure 4F). Results from both univariate and multivariate Cox regression analyses indicated that a higher pathomics score was an independent prognostic factor for KIRC (Figure 4G). Therefore, we believe that this pathomics model has potential clinical applications for assessing the prognostic risk of KIRC patients.

### 2.5. Impact of PYCRs Knockdown on In Vitro Cell Proliferation and Migration in Human Renal Cell Carcinoma Cell Line Caki-1 and A498 Cells

We validated the expression of PYCR1 and PYCR2 in normal human kidney tissue and renal cancer cell lines Caki-1, 786-O, and A498. The expression levels of PYCR1 and PYCR2 in renal cancer cells were significantly higher than those in normal kidney tissue (Figure 5A), consistent with the results of the previous analysis. We knocked down PYCR1 and PYCR2 in the human renal cell carcinoma cell lines Caki-1 and A498 cells to investigate their biological functions in KIRC cells (Figure 5B). The CCK-8 assay revealed that the growth rate of Caki-1 and A498 cells was inhibited after knocking down PYCR1 and PYCR2 (Figure 5C). EdU incorporation experiment results showed that knocking down PYCR1 and PYCR2 inhibited DNA replication in Caki-1 and A498 cells (Figure 5D). Colony formation assay results indicated that the growth rate of Caki-1 and A498 cells was inhibited after knocking down PYCR1 and PYCR2 (Figure 5E). Transwell migration assay demonstrated that the migration ability of Caki-1 and A498 cells decreased after knocking down PYCR1 and PYCR2 (Figure 5F). Similarly, the wound healing assay also indicated that knocking down PYCR2 inhibited the migration ability of Caki-1 and A498 cells (Figure 5G). We further observed the expression of epithelial-mesenchymal transition markers (Vimentin), proliferating cell nuclear antigen (PCNA), and epithelial calcium-binding protein (E-cadherin) in Caki-1 and A498 cells after PYCRs knockdown. The results showed that knocking down PYCR1 and PYCR2 downregulated the expression of Vimentin and PCNA while upregulating the expression of E-cadherin (Figure 5H). In conclusion, these results indicate that inhibiting PYCR1 and PYCR2 in vitro can suppress the growth and migration of renal cancer cells.

### 2.6. Impact of PYCRs Overexpression on In Vitro Cell Proliferation and Migration in Human Renal Cell Carcinoma Cell Line Caki-1 and 293T Cells

We investigated the biological effects of overexpressing PYCR1 and PYCR2 in the human renal cell carcinoma Caki-1 and human embryonic kidney 293T cells in vitro (Figure 6A). The CCK-8 assay results showed that overexpression of PYCR1 significantly promoted the growth rate of Caki-1 cells and 293T cells. However, overexpression of PYCR2 significantly accelerated the growth rate of Caki-1 cells while having no impact on the growth of 293T cells. (Figure 6B). EdU incorporation experiment confirmed that overexpressing PYCR1 accelerated DNA replication in Caki-1 and 293T cells. Additionally, PYCR2 was found to have an effect exclusively on Caki-1 cells. (Figure 6C). Colony formation assay indicated that PYCR1 overexpression led to increased growth of Caki-1 cells, and PYCR2 overexpression led to increased growth of 293T cells. (Figure 6D). Transwell migration assay showed that overexpression of PYCR1 and PYCR2 enhanced the migration ability of 293T cells, and overexpression of PYCR1 enhanced the migration ability of Caki-1 cells (Figure 6E). The wound healing assay also confirmed that PYCR1 overexpression accelerated the migration of Caki-1 and 293T cells, and PYCR2 overexpression accelerated the migration of 293T cells (Figure 6F). We further observed the expression of E-cadherin, N-cadherin, Vimentin, STAT3, Cyclin D1 and β-catenin in Caki-1 and 293T cells after overexpressing PYCRs. The results showed that overexpression of PYCR1 and PYCR2 in the Caki-1 cell line led to an upregulation of N-cadherin, Vimentin, and STAT3 while downregulating the expression of E-cadherin. In 293T cells, overexpression of PYCR1 and PYCR2 led to elevated levels of Cyclin D1 and N-cadherin while reducing E-cadherin expression (Figure 6G). Overall, these results suggest that overexpressing PYCR1 and PYCR2 in vitro can promote the growth and migration of renal cancer cells.

### 2.7. Regulation of the mTOR Signaling Pathway by PYCRs and Proline in Renal Cell Carcinoma

Given the impact of PYCR1 and PYCR2 knockdown and overexpression on the biological functions of renal cancer cells, we further investigated the potential molecular mechanisms involved in KIRC. We studied the influence of overexpressing PYCRs on the mTOR signaling pathway. The results showed that, compared to the control group, overexpression of PYCR1 and PYCR2 did not significantly alter the protein expression of mTOR and P-mTOR. However, downstream molecules of the mTOR signaling pathway, including p70S6K, P-p70S6K, and P-4EBP1, showed significant upregulation, while 4EBP1 expression was downregulated (Figure 7A). As PYCR is the final enzyme in the proline biosynthetic pathway, we further investigated the impact of different concentrations of proline on the mTOR signaling pathway. The results revealed that with increasing proline concentrations, the protein expression of mTOR and P-mTOR did not show significant changes. However, the expression of p70S6K, P-p70S6K, and P-4EBP1 was significantly upregulated, while 4EBP1 expression was downregulated (Figure 7B). These results suggest that PYCR1 and PYCR2 activate the mTOR signaling pathway by synthesizing proline, thereby influencing the proliferation and migration of renal cancer cells.

### 2.8. Impact and Mechanism Study of Downstream Pathways of PYCRs on the Survival of Renal Cancer Cells

Halofuginone (HF) is a prolyl-tRNA synthetase inhibitor, acting downstream of PYCR, preventing the insertion of proline into newly synthesized proteins, including proline-rich proteins such as collagen [22,23]. In this section, we used HF to inhibit the insertion of proline into newly synthesized proteins, blocking the synthesis of proline-containing proteins (such as collagen), and observed the impact of this inhibition on the proliferation and apoptosis of renal cancer cells. The results of the CCK-8 cytotoxicity assay showed that with increasing concentrations of HF, cell viability gradually decreased (Figure 7C). Colony formation experiments indicated that increasing HF concentrations inhibited the growth of Caki-1 cells (Figure 7D). CCK-8 cell proliferation assay results showed that with increasing HF concentrations, cell activity decreased (Figure 7E). We further observed the expression of E-cadherin, N-cadherin, PCNA, and Cyclin D1 in Caki-1 cells treated with different concentrations of HF. The results showed downregulation of N-cadherin, PCNA, and Cyclin D1 expression, while E-cadherin expression was upregulated (Figure 7F). At the same time, we detected the expression of apoptosis-related proteins BCL-2, BAX, BCL-XL, and PARP. The results showed downregulation of BCL-2, BCL-XL, and PARP expression, while BAX expression was upregulated (Figure 7G). These findings suggest that HF, by inhibiting the insertion of proline into newly synthesized proteins, exerts an inhibitory effect on the proliferation of renal cancer cells and may induce cell apoptosis by regulating the expression of related proteins. This provides an experimental basis for the potential application of HF in the treatment of renal cancer.

## 3. Discussion

The study of the PYCR gene family in pan-cancer has not been reported. In this research, we initially analyzed the expression patterns of PYCRs based on the TCGA database. Compared to adjacent tissues, PYCRs were upregulated in various tumors, and prognosis analysis indicated that PYCRs serve as adverse factors for overall survival (OS) in multiple cancers. This is consistent with previous studies reporting elevated expression of PYCR1 in cancers such as LIHC, BLCA, and KIRP, which can predict unfavorable prognosis outcomes [9,12,24]. Conversely, PYCR2, while less studied, shows promise as a prognostic biomarker in hepatocellular carcinoma and colorectal cancer [11,25], with PYCR3 having limited exploration primarily in nasopharyngeal cancer [26].

The emergence of immune checkpoint inhibitors (ICIs) represents a pivotal advancement in cancer therapy, yet resistance remains a critical challenge [24]. Identifying biomarkers predictive of ICI responsiveness is imperative, with both tumor mutational burden (TMB) and microsatellite instability (MSI) serving as established indicators [27,28,29,30,31]. Our analysis reveals a correlation between the expression of PYCRs and TMB and MSI in various tumors. Furthermore, elevated PYCR expression correlates positively with immune cell infiltration, implicating potential roles in tumor immune evasion [32,33]. The ability of PYCRs to modulate pathways such as the G2M checkpoint and E2F targets, essential for cell cycle regulation and tumor proliferation [34,35], contrasts sharply with their negative regulation of immune-related pathways like TNF-α signaling, NF-κB activation, and interferon-gamma response critical for anti-tumor immunity [36]. Therefore, the high expression of PYCRs in tumor tissues may assist immune escape by inhibiting inflammatory responses and anti-tumor immunity, thereby promoting tumor development.

Clear cell renal cell carcinoma (KIRC) is the predominant subtype of renal cell carcinoma [37]. In our pan-cancer analysis, we discovered that the PYCR gene family is highly expressed in all three types of kidney cancer. Consistent results were observed for PYCRs in various survival analyses (OS, PFS) for both KIRC and KIRP. Additionally, pathway enrichment analysis of PYCRs across all three types of kidney cancer also showed similar patterns. These findings have sparked significant research interest in the functions of PYCRs in kidney cancer. Our development of risk models based on PYCR expression profiles in KIRC highlights their potential clinical utility in predicting patient outcomes. High-risk groups exhibit enriched pathways such as E2F targets, G2/M checkpoints, and epithelial–mesenchymal transition (EMT), consistent with findings across various cancers [9,10,38,39].

Von Hippel-Lindau (VHL) is a tumor suppressor, and mutations in VHL occur in approximately 50% of KIRC patients [40]. In our analysis, the mutation rate of VHL in the high-risk score group was lower. Studies suggest ccRCC patients with intact VHL may respond better to immunotherapy [40]. These findings imply that high-risk KIRC patients might benefit from immunotherapy, possibly via CD4+ T cells-enhanced CTL activity [41]. These results suggest that patients in the high-risk score group in KIRC may have a better response to immunotherapy.

Precision medicine and genomic medicine combined with artificial intelligence have the potential to enhance patient healthcare [42]. AI, powered by vast datasets and advanced computing, has revolutionized cancer research by enhancing detection, classification, molecular profiling, drug discovery, and treatment outcome prediction [16]. Utilizing machine learning algorithms, we constructed robust pathomics feature models for accurate prognosis prediction in KIRC. This model, which utilizes high-dimensional histopathological data, enhances prognostic assessments and guides personalized treatment decisions. Our study underscores the potential of artificial intelligence to refine prognosis stratification and improve therapeutic outcomes in oncology.

Several studies have indicated that overexpression of PYCR1 and PYCR2 accelerates tumor growth [8,9,10,11,12]. In our experiments, silencing PYCR1 and PYCR2 in renal cancer cell line Caki-1 and A498 cells resulted in inhibited cell growth and migration, whereas overexpression of PYCR1 and PYCR2 promoted cell growth and migration. This aligns with research indicating PYCR1 in tumors such as colorectal cancer, bladder cancer, and gastric cancer, among others [43,44,45], and PYCR2 in colorectal cancer, hepatocellular carcinoma, melanoma, and other cancers [11,46,47]. Examining their impact on renal cancer cells, we focus on their regulation of the mTOR signaling pathway. Despite no significant changes in mTOR and P-mTOR protein expression, the upregulation of p70S6K, P-p70S6K, and P-4EBP1, with the downregulation of 4EBP1, suggested altered mTOR pathway activity. Proline metabolism, pivotal in cancer development, affects proliferation, invasion, and metastasis [48]. Varying proline concentrations showed similar effects, indicating that PYCR1 and PYCR2 activate mTOR via proline synthesis, influencing renal cancer cell proliferation and migration. These findings correlate with studies linking PYCR1 and PYCR2 to melanoma, hepatocellular carcinoma, and colorectal cancer proliferation via mTOR [15,39,49]. Halofuginone (HF), inhibiting prolyl-tRNA synthetase, blocks proline incorporation into proteins [22,23]. Cytotoxicity, colony formation assays, and cell proliferation assays suggest that HF suppresses renal cancer cell survival, growth, and proliferation. HF alters the expression of proliferation- and migration-related proteins and induces apoptosis, including inhibiting T-cell proliferation by blocking proline uptake [23].

Considering PYCR1, PYCR2, and HF effects on the mTOR pathway, these findings suggest targeting proline metabolism, especially via HF-mediated inhibition, as a potential therapeutic approach for renal cancer.

## 4. Materials and Methods

### 4.1. TCGA Pan-Cancer Data and Acquisition/Processing of H&E Pathology Images

RNA-Seq (RNASeqV2RSEM), clinical data, and mutation data for 33 cancers were obtained from the University of California, Santa Cruz (UCSC) Xena Browser (https://xena.ucsc.edu/, accessed date 17 July 2021). The R software (version 4.1.0; https://www.R-project.org) was employed to extract and integrate mRNA expression data of PYCRs and clinical data across the 33 cancers, followed by subsequent data analysis. Clinical data, corresponding sequencing data, and H&E pathology images of KIRC patients were downloaded from the TCGA database (https://portal.gdc.cancer.gov/, accessed date 11 May 2022).

### 4.2. Prognostic Analysis

We obtained prognostic information for TCGA tumor samples, including Overall Survival (OS) and Progression-Free Survival (PFS). The R package “survival” was employed for Cox analysis to calculate the correlation between gene expression levels and patient survival rates. The “forestplot” R package was used to create a forest plot. Kaplan–Meier (KM) method and log-rank tests were performed for survival analysis (*p* < 0.05), with the “survminer” and “survival” R packages used for plotting survival curves.

### 4.3. Immune Cell Infiltration Analysis and Gene Set Enrichment Analysis (GSEA)

The CIBERSORT algorithm was utilized to estimate the proportions of 22 immune cell infiltrations in samples from 33 cancer patients [50]. Visualization was conducted using the “ggplot2” R package. Gene Set Enrichment Analysis (GSEA) compared gene and functionally annotated gene sets based on the gene expression matrix, analyzing the enrichment differences between two groups of gene sets [51]. To explore the molecular mechanisms underlying prognostic differences between subgroups of the risk score model, we performed GSEA enrichment analysis on hallmark gene sets (h.all.v7.5.1.symbols.gmt) and Gene Ontology (GO) gene sets (c5.go.v7.4.symbols.gmt) using the “limma” and “clusterProfiler” R packages. The gene sets were downloaded from MSigDB (http://www.gsea-msigdb.org/gsea/msigdb, accessed date 11 May 2022). A *p*-value < 0.05 was considered statistically significant.

### 4.4. Tumor Mutation Burden (TMB) and Microsatellite Instability (MSI) Analysis

Tumor Mutation Burden (TMB), defined as the number of somatic mutations per megabase of the genomic sequence in the germline of a tumor, varies across malignancies [52]. We calculated the mutation count for 33 cancers from the somatic mutation dataset (MAF data), assessed TMB based on a Perl script, and normalized it by dividing it by the exon length. Microsatellite Instability (MSI) involves the spontaneous loss or gain of nucleotides in the repetitive DNA tract [53]. We obtained MSI scores for 33 cancers from the TCGA database. Spearman correlation analysis was conducted using the Spearman method to assess the correlation between PYCR expression and TMB or MSI. Visualization of the results was performed using the “fmsb” R package, generating a radar plot.

### 4.5. Construction of the KIRC Prognostic Risk Score Model

Using the “caret” R package, we randomly divided 507 TCGA-KIRC patients into training (*n* = 254) and internal validation (*n* = 253) cohorts (Appendix A). Leveraging the expression data of PYCR1 and PYCR2 in KIRC samples and overall survival (OS) data, a LASSO Cox regression analysis was conducted using the “glmnet” R package to construct a multi-gene risk score model. This model predicts the prognosis of KIRC patients, with the risk score established based on gene expression data and Cox regression coefficients, calculated by the formula: Risk Score = (0.050641 × Expression of PYCR1) + (0.051626 × Expression of PYCR2). The low-risk and high-risk groups were determined by the median of the risk scores. Cox regression coefficients and gene expression levels were utilized to compute the risk scores for each sample in the training and internal validation cohorts. Moreover, the external validation cohort dataset GSE167573 (*n* = 54) from GEO (https://www.ncbi.nlm.nih.gov/, accessed date 28 May 2024) was analyzed to validate the prognostic value of the risk score model in clear cell renal cell carcinoma. Based on the median risk score, KIRC samples were categorized into high-risk (training cohort *n* = 141, validation cohort *n* = 112) and low-risk groups (training cohort *n* = 113, validation cohort *n* = 141). The Kaplan-Meier method was employed to assess the overall survival differences between high- and low-risk groups. Additionally, the “timeROC” R package was used to draw time-dependent ROC curves and calculate the area under the curve (AUC) to validate the accuracy and predictive ability of the model. To validate the independent prognostic value of the risk score model and explore potential clinicopathological features related to prognosis, univariate and multivariate Cox regression analyses were conducted.

### 4.6. Pathological Image Segmentation and Feature Extraction

Pathological Image Acquisition: We downloaded pathological images from The Cancer Genome Atlas (TCGA) database (https://tcga-data.nci.nih.gov/tcga/, accessed date 14 November 2023). These images are tissue slices embedded in formalin and paraffin, with an SVS format and a maximum magnification of 20× or 40× (H&E stained pathological tissue images at 20× or 40× magnification). Pathologists reviewed the images, excluding sub-images with poor quality (contamination, blur, or blank areas exceeding 50%). The OTSU algorithm (https://opencv.org/) was employed to obtain the tissue regions of pathological slices. The 40× images were segmented into multiple 1024 × 1024 pixel sub-images, while the 20× images were segmented into multiple 512 × 512 pixel sub-images, which were then upsampled to 1024 × 1024 pixels. Ten random sub-images were selected from each pathological image for subsequent analysis. The PyRadiomics open-source package (https://pyradiomics.readthedocs.io/en/latest/, accessed date 1 July 2024) was utilized to standardize and extract features from the sub-images. After removing unnecessary image features, the mRMR and RFE algorithms were employed to select the optimal features, resulting in the final choice of 5 quantitative image features, which were Z-score standardized.

### 4.7. Pathomics Feature Model Construction

The GMB algorithm was employed to construct the pathomics feature model. The probability predicted by the pathomics feature model for the prognosis risk score model was designated as the pathomics_score. To evaluate the pathomics feature model, Receiver Operating Characteristic (ROC) curves and Precision-Recall (PR) curves were plotted. Diagnostic performance was assessed using Area Under the Curve (AUC) and other diagnostic indicators, including Brier score, recall, accuracy, sensitivity, specificity, Positive Predictive Value (PPV), and Negative Predictive Value (NPV). Calibration curves were used to demonstrate the calibration of the prediction model, and the goodness-of-fit was assessed using the Hosmer-Lemeshow test. Decision Curve Analysis (DCA) was employed to assess net benefit at different probability thresholds, evaluating clinical utility. Internal 5-fold cross-validation was conducted to validate the pathomics feature model. The sample cohort for building the prognosis model, after intersecting with KIRC samples with complete pathological image information, resulted in a total of 407 samples for pathological analysis. Using the ‘caret’ R package, these TCGA-KIRC 407 patients were randomly divided into training (*n* = 204) and validation (*n* = 203) cohorts (Appendix A), and the pathomics features of the training and validation cohorts were Z-score standardized.

### 4.8. Cell Culture and Transfection

Human renal cell carcinoma cell lines, namely Caki-1, 786-O, and A498, were purchased from the National Collection of Authenticated Cell Cultures of the Chinese Academy of Sciences and maintained in McCoy’s 5A, RPMI-1640 and EMEM (HyClone, Beijing, China) supplemented with 10% fetal bovine serum (Ausbian, Adelaide, Australia) at 37 °C with 5% CO_2_, respectively. Small interfering RNA (siRNA) targeting PYCR1 and PYCR2, along with the negative control (NC), were purchased from GenePharma (Shanghai, China). Transient transfection of PYCR1 or PYCR2 siRNA into Caki-1 and A498 cells was accomplished using Lipofectamine 3000 reagent (Invitrogen, Carlsbad, CA, USA) in OptiMEM^®^ I Reduced Serum Medium (OPTI-MEM) (Gibco, Thermo Fisher Scientific, Waltham, MA, USA). The siRNA sequences are detailed in Appendix A. PYCR1 and PYCR2 gene expression plasmids were constructed on the pBOB lentiviral vector and co-transfected with three viral packaging plasmids (pMDLg, pVSV-G, and pRSV-Rev) into 293T cells to generate complete infectious viral particles. In this study, lentivirus infection was employed in the establishment of gene overexpression models, and puromycin was utilized for the selection of stably expressing cells.

### 4.9. Protein Blotting

Proteins were extracted from cells using RIPA lysis buffer, which contained 1% phenyl methyl sulfonyl fluoride (PMSF, Sigma Aldrich, St. Louis, MO, USA), 1% sodium orthovanadate (Sigma Aldrich, St. Louis, MO, USA), and 1% aprotinin (Sigma Aldrich, St. Louis, MO, USA). The protein concentrations were measured by the BCA method. Protein samples (20 μg per well) were loaded onto 10% sodium dodecyl sulfate-polyacrylamide gel electrophoresis (SDS-PAGE). Subsequently, proteins were transferred to a polyvinylidene fluoride (PVDF) membrane (Millipore, Burlington, MA, USA). After blocking the membrane with 5% BSA at room temperature for 1 h, it was incubated overnight at 4 °C with primary antibodies. The membrane was then washed three times and incubated with secondary antibodies at room temperature for 1 h. Finally, protein bands were visualized using the ImageQuant LAS 4000mini gel imaging system (General Electric Company, Boston, MA, USA). The primary antibodies used in the immunoblotting studies were as follows: β-actin, GAPDH, α-tubulin, PYCR1, PYCR2, Vimentin, Bax, Bcl2 (#Cat:20536-1-AP, 10494-1-AP, 11224-1-AP, 13108-1-AP, 55060-1-AP, 10366-1-AP, 50599-2-Ig, 50599-2-Ig, 68103-1-Ig, Proteintech Group, Wuhan, China), N-Cadherin, E-Cadherin (#PTM-5221, PTM-6222, PTM biolabs, Hangzhou, China), CyclinD1, PCNA, Stat3, PARP, Bcl-xl, mTOR, Phospho-mTOR (Ser2448), 4E-BP1, Phospho-4E-BP1 (Thr37/46), p70 S6 Kinase, Phospho-p70 S6 Kinase (Thr389), β-Catenin (#55506, #13110, #9139, #9532, #2764, #2972, #2971, #9644, #2855, #9202, #9205, #8480, Cell Signaling Technology, Danvers, MA, USA).

### 4.10. Cell Proliferation

Cell viability was assessed using the CCK-8 assay. Briefly, 2000 cells were seeded in a 96-well plate and cultured according to the recommended instructions. Then, 10 μL of CCK-8 reagent (Zomanbio, Beijing, China) was added to each well. After incubating for an additional 2 h, the absorbance was measured at 450 nm using a microplate reader (Thermo Fisher Scientific, Waltham, MA, USA). EdU incorporation assay was performed to detect cell proliferation. Caki-1 and A498 cells (1 × 10^5^) were seeded in a 24-well plate, transfected with PYCRs siRNA or control siRNA for 48 h, and then treated with 10 µM EdU for an additional 1.5 h. Cells were fixed with 4% paraformaldehyde and stained with Azide 488 and Hoechst 33342 (Beyotime, Shanghai, China), followed by imaging using a fluorescence microscope.

### 4.11. Cell Migration

Wound Healing Assay: Cells were seeded in a 6-well plate and allowed to reach approximately 90% confluence. Using a sterile 200 μL pipette tip, a wound was created on the cell monolayer. The cells were then cultured in basal medium, and photographs were taken at 0 and 24 h under a microscope to record the wound width. The cell migration rate was calculated using the formula: (Width at 0 h − Width at 24 h)/Width at 0 h. Transwell Assay: Caki-1 cells (6 × 10^4^) and A498 cells (2 × 10^4^) were seeded in the upper chamber of Transwell inserts (8-µm pore size; Corning, NY, USA). McCoy’s 5A/EMEM containing 20% FBS was added to the lower chamber. After 24 h of incubation at 37 °C and 5% CO_2_, the migrated cells in the lower chamber were fixed with 4% paraformaldehyde and stained with crystal violet. Photographs of cells in five random fields under a microscope (10×) were taken, and the relative area ratio was calculated.

### 4.12. Drug Treatment

The cytotoxicity of halofuginone (HF) (Sigma Aldrich, St. Louis, MO, USA) on Caki-1 cells was determined by the CCK8 method. Briefly, a total of 5000 cells were seeded in 96-well plates. Caki-1 cells were exposed to halofuginone of a variety of concentrations (0, 50, 100, 200, 400 nM) for 12, 24, 36, and 48 h. Then, the cell viabilities were measured by the CCK8 method. In colony formation assay, cells were pre-treated with different concentrations of HF (0, 50, 100, 200 nM) for 24 h. Then, cells were seeded into a 12-well plate with 5000 cells per well. Cells were cultured for 7 days and then fixed with 4% paraformaldehyde and stained with 1% crystal violet. Colony-forming ability was observed through both naked-eye examination and microscopy. In a separate experiment, Caki-1 cells were exposed to diverse HF concentrations (0, 50, 100, 200, 400 nM) for 24 h, and cellular proteins were extracted for subsequent Western blot analysis. Caki-1 cells were exposed to diverse L-proline (Sigma Aldrich, St. Louis, MO, USA) concentrations (0,1, 2, 4, 8, 16 mM) for 24 h, and cellular proteins were extracted for subsequent Western blot analysis.

### 4.13. Statistical Analysis

Statistical analysis was performed using R language (version 4.1.0; https://www.R-project.org) and GraphPad Prism 8.0 (GraphPad Software, San Diego, CA, USA). Wilcoxon signed-rank test was employed to compare the expression of PYCRs mRNA between tumor tissues and normal tissues. One-way analysis of variance (ANOVA) was used for comparisons among multiple groups. Spearman correlation analysis was applied to explore the correlation between continuous variables. Single-factor Cox proportional hazards regression model or log-rank test was used to assess the correlation between gene expression and the overall survival rate of patients. A significance level of *p* < 0.05 was considered statistically significant.

## 5. Conclusions

In summary, our study highlights PYCRs as significant prognostic biomarkers across diverse cancers, influencing tumor biology and immune responses. Targeting PYCR-mediated pathways, particularly in the context of immune checkpoint therapy and precision medicine, holds promise for improving clinical outcomes in cancer patients. Further investigation into PYCR function and therapeutic targeting is warranted to fully exploit their potential in oncology.

## Figures and Tables

**Figure 1 ijms-25-08096-f001:**
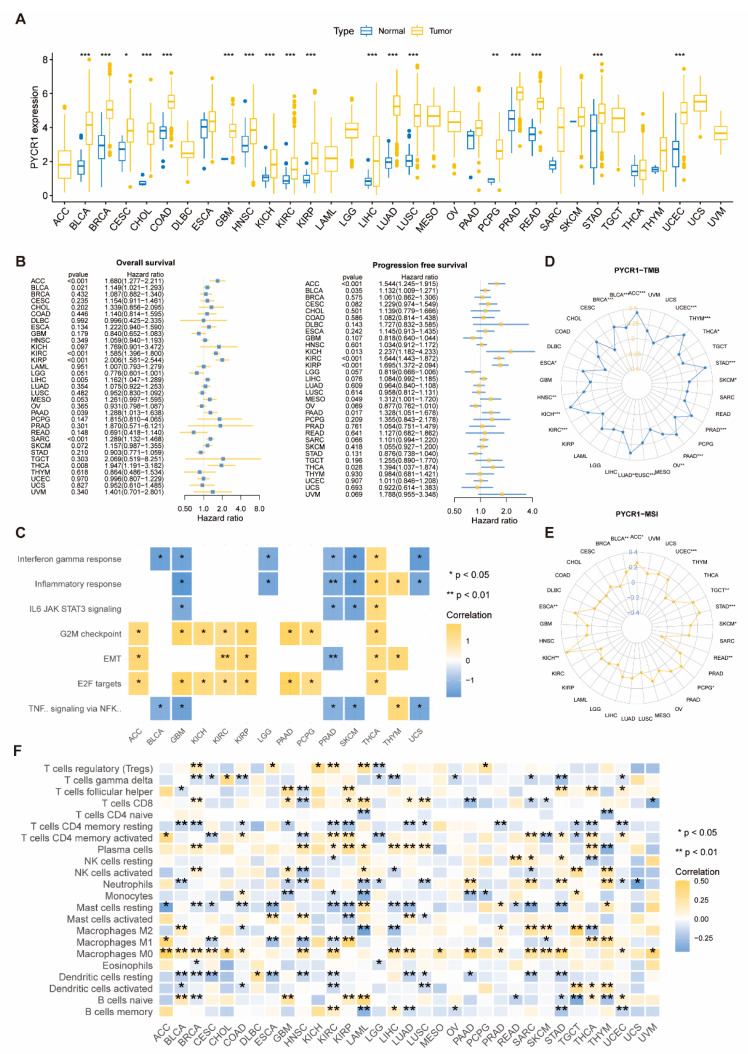
**Analysis of PYCR1 in Pan-cancer.** (**A**) Differential expression analysis of PYCR1; (**B**) results of single-factor Cox regression analysis of PYCR1 with prognosis (OS and PFS); (**C**) visualization of seven pathways enriched by PYCR1 in multiple tumors; (**D**,**E**) analysis of the expression of PYCR1 with TMB and MSI; (**F**) analysis of the expression of PYCR1 with the infiltration of 22 immune cells; * *p* < 0.05, ** *p* < 0.01, and *** *p* < 0.001.

**Figure 2 ijms-25-08096-f002:**
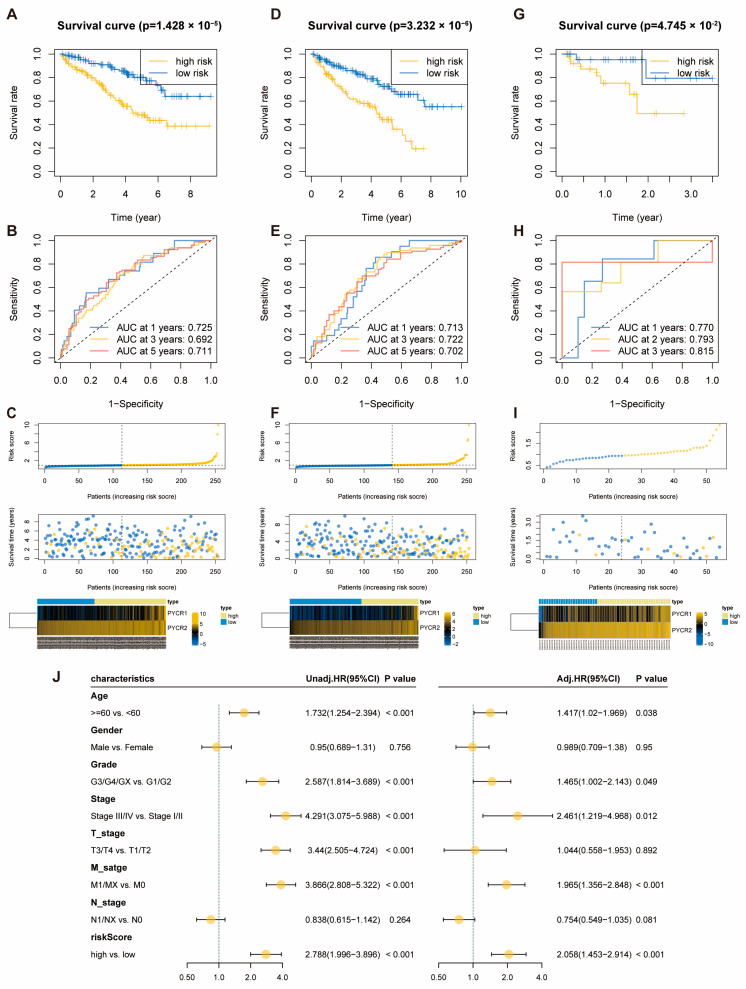
**Relationship between KIRC Risk Score Model and Patient Prognosis.** (**A**,**D**,**G**) Kaplan–Meier curves for high and low-risk score groups in the training, the internal validation and the external validation cohorts; (**B**,**E**,**H**) time-dependent ROC curves for the training, the internal validation and the external validation cohorts; (**C**,**F**,**I**) patient risk score distribution, scatter plot of patient survival status, and expression patterns of risk genes in training, the internal validation and the external validation cohorts from top to bottom; (**J**) univariate and multivariate Cox regression analysis of the prognosis risk score model.

**Figure 3 ijms-25-08096-f003:**
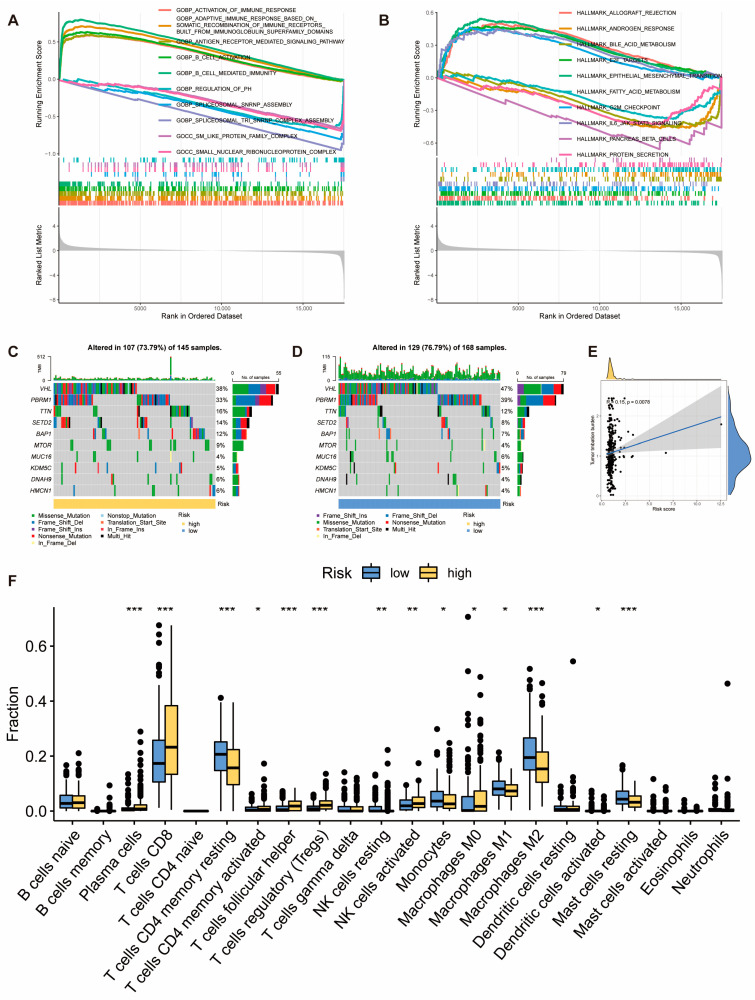
**Molecular and Immune Features of Subgroups in the KIRC Risk Score Model.** (**A**) GSEA enrichment analysis of the KIRC risk score model in the GO gene set (*p* < 0.05); (**B**) GSEA enrichment analysis of the KIRC risk score model in the hallmark gene set (*p* < 0.05); (**C**,**D**) gene mutation spectra of patients in the high and low-risk groups of KIRC; (**E**) correlation analysis between tumor mutation burden (TMB) and risk score; (**F**) box plots comparing the infiltration abundance of 22 immune cells in the subgroups of the risk score model in KIRC. * *p* < 0.05, ** *p* < 0.01, and *** *p* < 0.001.

**Figure 4 ijms-25-08096-f004:**
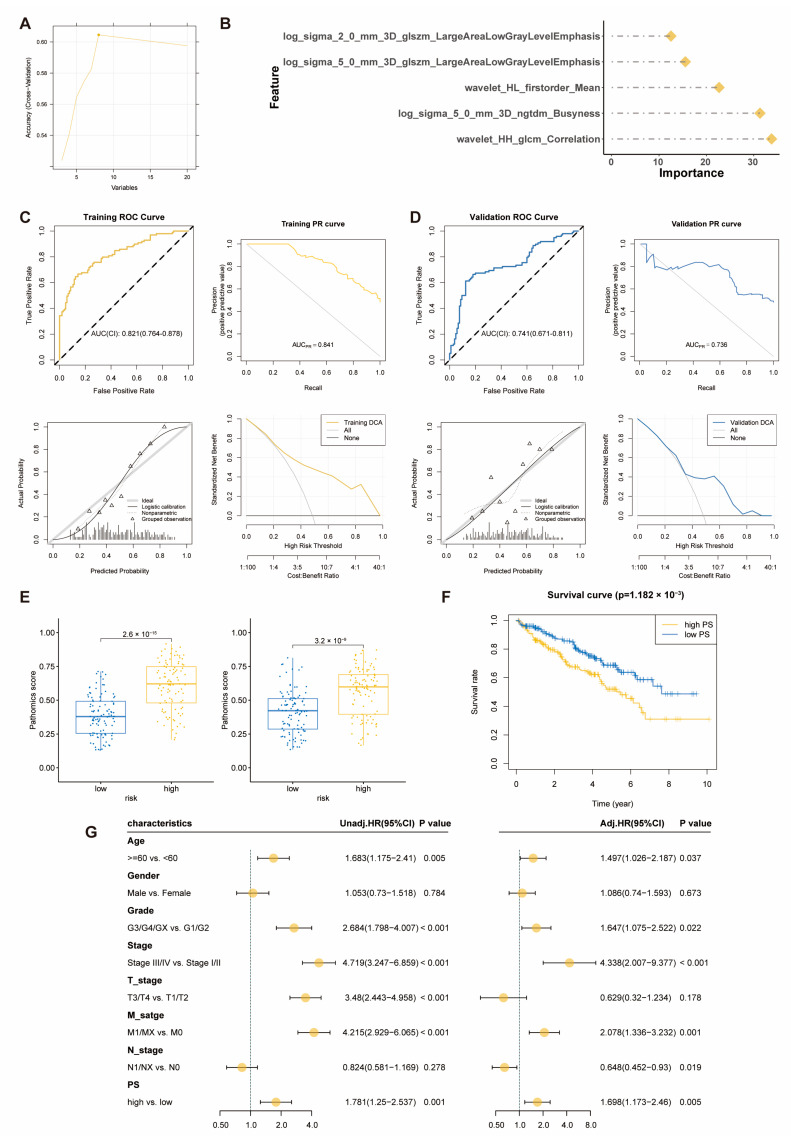
**Pathomic Feature Model and Prognostic Analysis in KIRC.** (**A**) Pathomics feature selection; (**B**) establishment of the GBM model; (**C**) performance evaluation of the model in the training cohort; (**D**) performance evaluation of the model in the validation cohort; (**E**) distribution of pathomics score values in the high and low-risk groups of the prognostic risk score model in the training and validation cohorts; (**F**) Kaplan–Meier curve for the pathomics feature model and OS; (**G**): Univariate and multivariate Cox regression analysis of the pathomics feature model.

**Figure 5 ijms-25-08096-f005:**
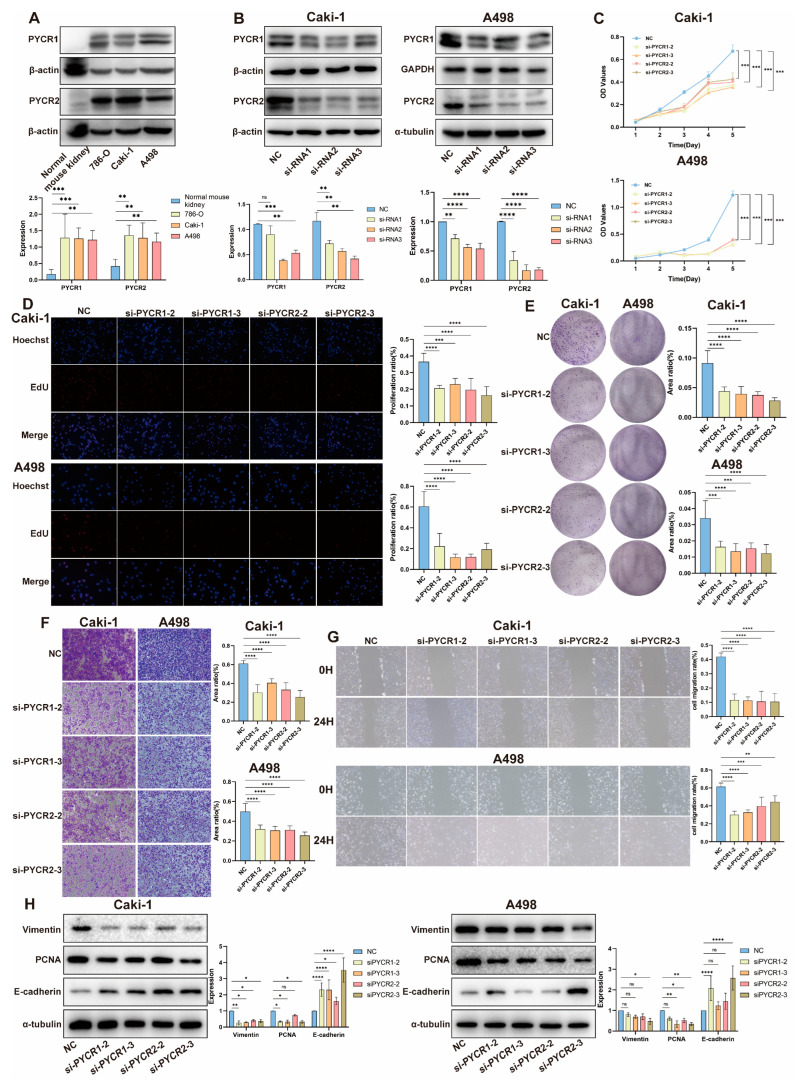
**Biological Functional Validation of PYCRs Knockdown in Renal Cancer Cell Lines.** (**A**) Validation of PYCRs expression; (**B**) validation of silenced PYCRs expression; (**C**) CCK8 viability assay; (**D**) EdU incorporation experiment; (**E**) colony formation assay; (**F**) transwell migration assay (**G**) scratch healing experiment; (**H**) assessment of the impact of silenced PYCRs on the expression of proliferation and migration-related proteins. ns, *p* > 0.05, * *p* < 0.05, ** *p* < 0.01, *** *p* < 0.001, **** *p* < 0.0001.

**Figure 6 ijms-25-08096-f006:**
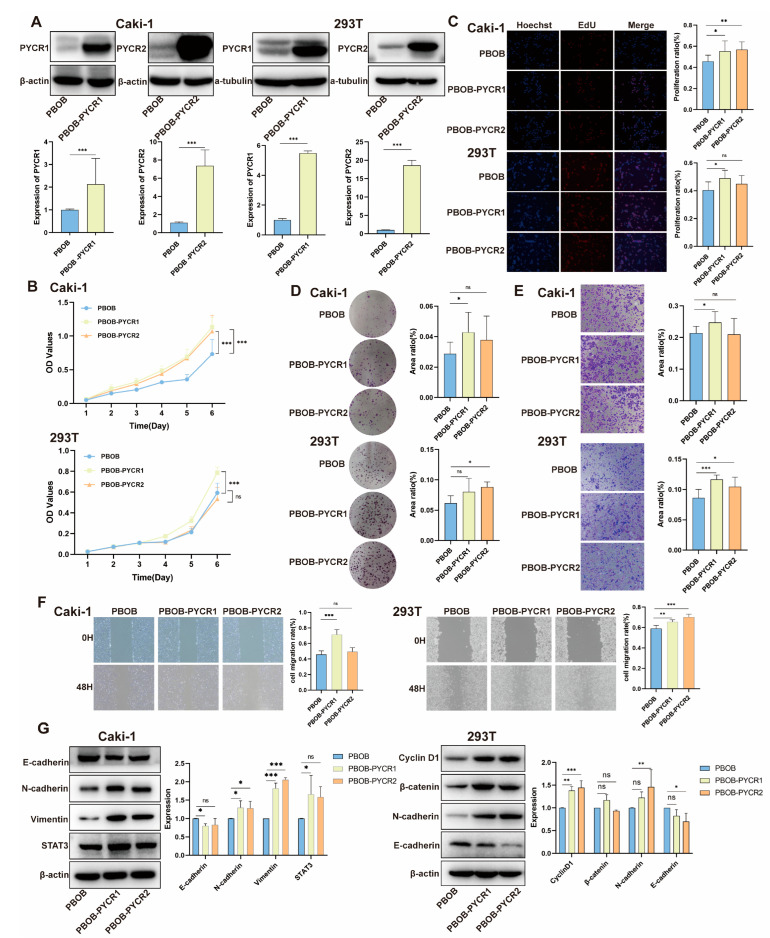
**Biological Functional Validation of PYCRs Overexpression in Renal Cancer Cell Lines.** (**A**) Validation of PYCRs overexpression; (**B**) CCK8 viability assay; (**C**) EdU incorporation experiment; (**D**) colony formation assay; (**E**) transwell migration assay; (**F**) scratch healing experiment; (**G**) assessment of the impact of PYCRs overexpression on the expression of proliferation and migration-related proteins. ns, *p* > 0.05, * *p* < 0.05, ** *p* < 0.01, *** *p* < 0.001.

**Figure 7 ijms-25-08096-f007:**
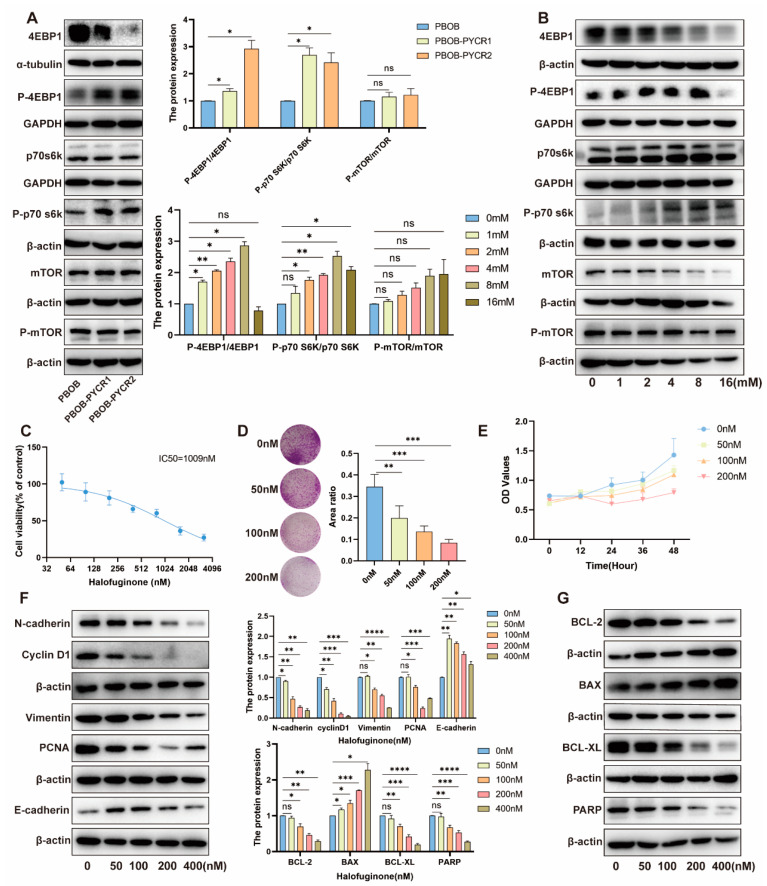
**Exploration of the Regulatory Mechanisms of PYCRs in Renal Cancer Cell Lines.** (**A**) Assessment of the impact of PYCRs overexpression on the mTOR signaling pathway in renal cancer cells; (**B**) influence of proline treatment on the mTOR signaling pathway in renal cancer cells; (**C**) CCK8 cytotoxicity assay; (**D**) colony formation assay; (**E**) CCK8 viability assay; (**F**) evaluation of the expression of proliferation and migration-related proteins in renal cancer cells treated with Halofuginone (HF); (**G**) detection of apoptosis-related protein expression in renal cancer cells treated with HF. ns, *p* > 0.05, * *p* < 0.05, ** *p* < 0.01, *** *p* < 0.001, **** *p* < 0.0001.

## Data Availability

Publicly available datasets were analyzed in this study. These data can be found here: 33 types of cancers from the Xena browser (https://xena.ucsc.edu/, accessed date 17 July 2021) and the external validation cohort from the GEO database (https://www.ncbi.nlm.nih.gov/geo/, accessed date 28 May 2024).

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
