# Peer review of "Deciphering the Effects of the PYCR Family on Cell Function, Prognostic Value, Immune Infiltration in ccRCC and Pan-Cancer"

_ijms, 2024, doi:10.3390/ijms25158096_

Round 1

Reviewer 1 Report (Previous Reviewer 2)

Comments and Suggestions for Authors

The authors have done a remarkable job in addressing my comments. I now support publication of this study.

Author Response

Comment 1::The authors have done a remarkable job in addressing my comments. I now support publication of this study.

Response 1: Thank you for your recognition of our work. We are very pleased to hear that you now support the publication of this study. We appreciate your valuable comments and suggestions during the review process.

Reviewer 2 Report (Previous Reviewer 1)

Comments and Suggestions for Authors

The blots of figure 5 panel A,B are too impressed, if the blot is too exposed the differences are not appreciated, for example in actin and in the tubulin, invalidating the quantification. For the same reason in figure 6G the Ecadherin/N cadherin switch is not well observed.

 In general the figures in figure 5 should be enlarged, so as to better appreciate the images related to colony formation, migration, and the incorporation of edu

Author Response

Comment 1::The blots of figure 5 panel A,B are too impressed, if the blot is too exposed the differences are not appreciated, for example in actin and in the tubulin, invalidating the quantification. For the same reason in figure 6G the Ecadherin/N cadherin switch is not well observed.

Response 1:Replaced the overexposed images in Figures 5A, 5B, and 6G with WB images of the same samples but with shorter exposure times.

Comment 2:: In general the figures in figure 5 should be enlarged, so as to better appreciate the images related to colony formation, migration, and the incorporation of edu

Response 2: Enlarged the images of the colony formation assay, EdU incorporation assay, and Transwell assay in Figure 5.

This manuscript is a resubmission of an earlier submission. The following is a list of the peer review reports and author responses from that submission.

Round 1

Reviewer 1 Report

Comments and Suggestions for Authors

In general this work displays an interesting example of informatic tools combined with classical analysis and the scientific value that can be achieved. The work is well done and very interesting, but I suggest to insert a graph on the activity of the interested pathways (PYCR and downstream) and a flow map that represents graphically the sequence of application of all the used computer tools, the purpose and subsequent conclusions.

Line 54"certain cancers" Please provide a list of examples

Line 57-64: Very interesting and well written part, I do not suggest big changes , but please enter clearly the model of tumor referred to( instead of the name of the cell line) and whether the evidence is collected in vivo or in vitro.

68-74: This section contains some redundant sentences, please rewrite.

Materials and methos: Please add in each subsection the number of seeded cells (or the amount of proteins)

Wound healing assay: In the study of wound healing by scratch assay it is necessary to block the cellular proliferation, which in the chosen models is very rapid, and in general can be induced by mechanical stimuli such as the increase of available space (scratch). Therefore, if the authors want to analyze and consider this data as migration index, it is necessary that there is the addition of an antimitotic (for example Ara-C). If the authors have used an antimitotic please indicate what and in what dose, otherwise re-evaluate the data as a combined, proliferative and migratory action, especially at prolonged times.

DRUG: the action and the complete name of HF must be indicated the first time the abbreviation appears.

Figures: I encurage the authors to increase the size of the graphs and the presented figures, to increase the quality and the possibility to appreciate the data.

Western blot data: unfortunately the blots presented are poor in quality, in fact blots too impressed (with too black bands) prevent you from evaluating and observing the modulation of protein expression. In addition, the actins are poorly balanced. All this interferes with the quantification that is to be reviewed. Please replace blots, increase their quality and review the quantification/statistical analysis accordingly.

Figure legends: please provide a description of comparison term for p value (vs what is it calculated?)

If the authors have the chances, this paper may benefit from the analysisi of some stemness markers or biological functions, linked to cancer aggressivness and progression

Discussion: Please reduce, it's too long and many parts are redundant. 

My overall opinion is positive and I'm sure the authors can provide an improved version of this paper.

Reviewer 2 Report

Comments and Suggestions for Authors

In "Deciphering the effects of PYCR family on cell function, prognostic value, immune infiltration in ccRCC and pan-cancer", Chen and coworkers set out to do bio-informatic analysis on the role of the three PYCR family genes in cancer, then perform in vitro validation experiments in ccRCC. This is a potentially very interesting study.

The authors do an admirable job of data-mining the TCGA collection of cancer genomics and transcriptomics, and find interesting correlations between PYCR gene expression and clinical parameters, and suggest usefulness of PYCR as biomarker genes. Their follow up in kidney cancer (ccRCC) leaves a lot to be desired though, and prevents me from accepting the manuscript in its current form.

My concerns are:

Major:

1. The authors need to provide a rationale why kidney cancer was chosen for follow-up experiments, and none of the other cancer types that had interesting PYCR-clinical correlations.

2. The authors start in vitro validation experiments with 3 ccRCC cell lines, but apart from Figure 5A show only data for CAKI-1. All validation experiments need to be in a minimum of 2 different cancer cell lines, since tens to hundreds of non-associated other gene mutations in these cell lines can lead to singular molecular pathway peculiarities.

3. The authors use only 1 siRNA sequence per gene. A minimum of 2 siRNA sequences should be used to prevent the influence of off-target events.

4. The authors suggest the use of PYCR gene expression as a clinical cancer biomarker. This is not very useful without showing data for a second (validation) cancer cohort, and without comparison with existing biomarkers - or showing that no such markers currently exist. 

Minor:

1. The use of normal kidney tissue versus ccRCC cell lines is not sufficient for the conclusion that PYCR genes are over-expressed in cancer (Figure 5A). 

2. The scratch assay pictures in Figure 5G and 6F are not consistent: at the 0 h timepoint scratches have different widths.

3. The authors suggest the use of HF as a clinical intervention. They should mention whether HF or a similar compound is useful in vivo, is being tested in clinical trials and/or is available in GMP-grade.